# MVMRmode: Introducing an R package for plurality valid estimators for multivariable Mendelian randomisation

**Benjamin Woolf**[1,2,3]*, **Dipender Gill**[4], **Andrew J. Grant**[5], **Stephen Burgess**[3]

1 School of Psychological Science, University of Bristol, Bristol, United Kingdom, 2 MRC Integrative Epidemiology Unit, University of Bristol, Bristol, United Kingdom, 3 MRC Biostatistics Unit, University of Cambridge, Cambridge, United Kingdom, 4 Department of Epidemiology and Biostatistics, School of Public Health, Imperial College London, London, United Kingdom, 5 Sydney School of Public Health, Faculty of Medicine and Health, The University of Sydney, Sydney, NSW, Australia

* benjamin.woolf@bristol.ac.uk

## Abstract

### Background

Mendelian randomisation (MR) is the use of genetic variants as instrumental variables. Mode-based estimators (MBE) are one of the most popular types of estimators used in uni-variable-MR studies and is often used as a sensitivity analysis for pleiotropy. However, because there are no plurality valid regression estimators, modal estimators for multivariable-MR have been under-explored.

### Methods

We use the residual framework for multivariable-MR to introduce two multivariable modal estimators: multivariable-MBE, which uses IVW to create residuals fed into a traditional plurality valid estimator, and an estimator which instead has the residuals fed into the contamination mixture method (CM), multivariable-CM. We then use Monte-Carlo simulations to explore the performance of these estimators when compared to existing ones and re-analyse the data used by Grant and Burgess (2021) looking at the causal effect of intelligence, education, and household income on Alzheimer's disease as an applied example.

### Results

In our simulation, we found that multivariable-MBE was generally too variable to be much use. Multivariable-CM produced more precise estimates on the other hand. Multivariable-CM performed better than MR-Egger in almost all settings, and Weighted Median under balanced pleiotropy. However, it underperformed Weighted Median when there was a moderate amount of directional pleiotropy. Our re-analysis supported the conclusion of Grant and Burgess (2021), that intelligence had a protective effect on Alzheimer's disease, while education, and household income do not have a causal effect.

**Data Availability Statement:** All data produced in the present study are available from DOI 10.17605/OSF.IO/8DZKU.

**Funding:** Benjamin Woolf is funded by an Economic and Social Research Council (ESRC) South West Doctoral Training Partnership (SWDTP) 1+3 PhD Studentship Award (ES/P000630/1). The funders had no role in study design, data collection and analysis, decision to publish, or preparation of the manuscript.

**Competing interests:** The authors have declared that no competing interests exist.

## Conclusions

Here we introduced two, non-regression-based, plurality valid estimators for multivariable MR. Of these, "multivariable-CM" which uses IVW to create residuals fed into a contamina-tion-mixture model, performed the best. This estimator uses a plurality of variants valid assumption, and appears to provide precise and unbiased estimates in the presence of balanced pleiotropy and small amounts of directional pleiotropy.

## Background

Mendelian randomisation (MR) is an increasingly popular method for causal inference in epidemiology which uses the random assignment of genetic variants at birth to justify the assumptions of an Instrumental variables analysis [1, 2]. In a traditional MR study, genetic variants (typically single-nucleotide polymorphisms, SNPs) which robustly associate (typically at genome-wide significance) with an exposure of interest are selected as instruments [3]. Because of the easy accessibility of Genome-Wide Association Study (GWAS) summary statistics for many epidemiological traits, MR is often implemented using summary data, in a so-called 'two-sample MR' analysis [4]. In such a setting, the effect of the exposure on the outcome is estimated using a Wald ratio as the variant-outcome association divided by the genotype-exposure association. When there are multiple variants, their effects are generally combined using an inverse variance weighted (IVW) meta-analysis.

On top of requiring a robust genotype-exposure association, instrumental variables analysis requires that there are no variant-outcome confounders, and that the variant can only cause the outcome via the exposure. The first of these assumptions is justified by Mendel's laws of independent and random segregation. However, the second assumption is less plausible due to pleiotropy (the association of most variants with multiple traits). Pleiotropy can occur for two reasons: Firstly, if the exposure causes many other traits, then the genetic variants which associate with it should also associate with these other traits. This type of pleiotropy (often called vertical pleiotropy) is required for MR to work. However, the second type of pleiotropy (horizontal pleiotropy) occurs when the genetic variants independently cause two phenotypes. A second advantage of two-sample MR is that it allows for the implementation of 'pleiotropy robust' estimators [5]. These methods generally allow for some variants to be pleiotropic by modifying the assumptions of the instrumental variables framework. One of the first methods proposed for doing this is MR-Egger. IVW can be conceptualised as a weighted intercept-free regression of the variant-outcome associations on the variant-exposure associations. MR-Egger fits the same model as IVW but with an intercept. This model is robust to pleiotropy if the instrument strength is independent of the strength of the direct, pleiotropic, effect (called the InSIDE assumption) [6].

A recent systematic review of two-sample MR studies found that the most frequently implemented pleiotropy robust estimators were MR-Egger, weighted median, and weighted mode [7]. Weighted Median will provide valid estimates if at least half the variants are valid instruments, and so is called a 'majority valid' estimator. Weighted mode makes the ZEro Modal Pleiotropy Assumption (ZEMPA), i.e. that there is zero pleiotropy in the modal estimand of the causal effect [8]. ZEMPA is plausible because we should expect the causal effects for variants which are valid instruments to be similar, but each invalid variant to have its own unique pleiotropic bias [9]. If the unique paths are independent of each other, then so too should the biases they exert on invalid variants. Thus, valid variants should have clustered effect estimates,

while invalid variants should create heterogeneity. Hence, in settings where there are some valid instruments, we should expect the most common effect estimated to be the valid causal parameter. Here in, we call this type of estimator, which will produce valid estimates when a plurality of SNPs are valid, 'plurality valid' estimators.

Estimating modes directly from observed data can be difficult because no two estimates are ever exactly equal. Therefore, the most common observation at a given level of precision may be very different from the true mode. Traditional MBEs avoid this dilemma by smoothing the observed distribution using a parametric kernel-density-smoothed function. This converts the observed estimates into a probability density distribution, and then select the mode of this distribution. An alternative plurality valid estimator comes from the contamination mixture method [10].

The contamination mixture method uses a maximum likelihood approach, assuming the variant specific Wald ratios are normally distributed [10]. It produces a consistent estimator of the causal effect under the plurality valid (ZEMPA) assumption. The advantages of the contamination mixture method are that it does not require the parametric assumptions of the kernel-density function, is more computationally efficient, and generally produces more precise estimates with potentially asymmetric confidence intervals [10].

Multivariable MR (MVMR) is an extension of MR to allow for the simultaneous modelling of the effect of multiple exposures on an outcome [11]. The effects of each exposure in an MVMR model are the direct effects of the exposure on the outcome conditional on the other exposures. This has resulted in MVMR being applied as a method for mediation analyses [12], but it is also used to adjust for known biases in an MR model [13–15]. MVMR modifies the three instrumental Variables assumptions so that the variant is a valid instrument if: 1) the variant is robustly associated with at least one exposure, 2) there are no variant-outcome confounders, 3) the variant can only cause the outcome via one or more of the exposures.

MVMR was originally introduced using a residual-based framework, in which the effect of a second exposure on the outcome was removed from the variant-outcome association, and the effect of the second exposure on the exposure was removed from the variant-exposure association [14]. These modified associations were then used as the input to a traditional MR estimator. However, given the analogy between IVW and weighted regression, two-sample MVMR is typically implemented as a type of multiple regression, in which the variant-outcome associations for the variants which associate with either exposure of interest are regressed on the variant-exposure associations in an intercept-free linear regression, inversely weighted by the variance in the variant-outcome association. MR-Egger can also be implemented by allowing for a non-zero regression intercept, and weighted median can be implemented using weighted quantile regression [16].

However, we are not aware of an existing estimators for doing mode-based regression, and hence MVMR which make a plurality valid-type assumption like ZEMPA have been underexplored. The multivariable constrained maximum likelihood (MVMR-cML) method provides consistent estimates under a plurality-valid assumption by maximizing a constrained likelihood function subject a maximum number of invalid instruments [17]. The MVMR-Horse method provides estimates under the same model as MVMR-cML in a Bayesian framework, using horseshoe priors for identification [18]. Finally, the Genome-wide mR Analysis under Pervasive PLEiotropy (GRAPPLE) method is a multivariable method that can provide robust estimates in the presence of invalid instruments using profile likelihood [19]. Here, we, introduce and validate a further framework for implementing plurality valid estimators in two-sample MVMR.

## Methods

### Theoretical background

**Notation and assumptions.** We assume a set of genetic variants that are independently distributed are being proposed as instruments in an MR analysis. We shall denote with subscript i the ith element of any vector, which relates to the ith genetic variant. Let $\beta_{y,i}$ be the genetic variant-outcome association for the ith genetic variant and $\beta_{x,i}$ be the genetic variant-exposure association for the ith variant. We represent the causal effect of the exposure on the outcome using the scalar $\theta$. We also assume that the exposure-outcome relationship is linear and unaffected by effect modification. We let $\alpha_i$ represent pleiotropic effects of the ith variant on the outcome. Thus, when $\alpha_i = 0$, the $i$th variant is a valid instrument.

Suppose the ith variant-exposure and variant-outcome associations are related according to the model proposed by Bowden et al. [20]:

$$\beta_{y,i} = \theta\beta_{x,i} + \alpha_i \qquad\qquad 1)$$

Now suppose we have estimates for two exposures, denoted by $x_1$ and $x_2$. $\beta_{x_1,i}$ and $\beta_{x_2,i}$ are the ith variant's associations with the first and second exposure, respectively. Likewise, $\theta_1$ and $\theta_2$ are the causal effects of the first and second exposure, respectively, on the outcome. We can now extend (1) as follows:

$$\beta_{y,i} = \theta_1\beta_{x_1,i} + \theta_2\beta_{x_2,i} + \alpha_i' \qquad\qquad 2)$$

Where $\alpha_i'$ represents pleiotropic effects of the ith variant on the outcome which do not pass via $x_1$ or $x_2$.

**Statistical framework.** In practice, we do not observe $\beta_y$, $\beta_{x_1}$, or $\beta_{x_2}$. However, we may obtain estimates, for example from GWAS. We denote the vectors of association estimates by $\hat{\beta}_y$ $\hat{\beta}_{x_1}$, and $\hat{\beta}_{x_2}$. Thus, in traditional multivariable-IVW we can estimate $\theta_1$ and $\theta_2$ using the following linear model:

$$\hat{\beta}_y = \theta_1\hat{\beta}_{x_1} + \theta_2\hat{\beta}_{x_2} + \varepsilon_1; \text{ and } \varepsilon_{1,i} \sim N(0, \sigma_{y,i}^2). \qquad\qquad 3)$$

Given the data structure in Eqs (2), (3) will provide a consistent estimator when $\alpha_i' = 0$ for all i (i.e., all variants are valid instruments), or when $\sum_1^n \alpha_i' = 0$ and $\alpha_i'$ is independent of $\hat{\beta}_{x_{1,i}}$ and $\hat{\beta}_{x_{2,i}}$ for all i (i.e., pleiotropy is balanced and the InSIDE assumption is met). A plurality valid estimator, on the other hand, should be consistent provided that a plurality of the $\alpha_i'$ are zero, i.e. under the ZEMPA assumption.

Let $\tilde{\beta}_y$ be the residuals from regressing $\hat{\beta}_y$ on $\hat{\beta}_{x_2}$ (without an intercept), and let $\tilde{\beta}_{x1}$ be the residuals from regressing $\hat{\beta}_{x_1}$ on $\hat{\beta}_{x_2}$ (without an intercept). We can now estimate $\theta_1$ using the linear model:

$$\tilde{\beta}_y = \theta_1\tilde{\beta}_{x1} + \varepsilon_2; \text{ and } \varepsilon_{2,i} \sim N(0, \sigma_y^2). \qquad\qquad 4)$$

Let $\tilde{\alpha}$ be the residuals from regressing a vector of the pleiotropic effects on $\hat{\beta}_{x_2}$ (without an intercept). Because we have now reformulated the equation for the variant-outcome association so that it is in terms of a univariable regression model, $\tilde{\beta}_y$ and $\tilde{\beta}_{x1}$ can be used as the inputs to a traditional univariable mode-based estimator. When more than one exposure is of interest, then this process can be iterated for each exposure. It follows that a plurality valid estimator for $\theta_1$ using the residuals in this way will produce a valid estimate

provided that a plurality of the $\tilde{\alpha}_i$ values are zero. This seems likely to be the case if a plurality of the $\alpha_i'$ values are zero and the non-zero elements are distributed around zero (i.e., balanced pleiotropy).

In settings with only two exposures, the residuals could be obtained through univariable MR of the outcome on the second exposure, and of the first exposure on the second exposure. Where there are more than two exposures, an existing multivariable MR method could be used instead to create residuals. This general framework could be implemented using a variety of estimators. Here we explore two types of plurality valid estimators. Firstly, we explore an estimator which uses a regression model to create the residuals fed into a traditional mode-based estimator (MBE) [8], which we dub 'multivariable-MBE'. This regression model could be created using any of the existing MVMR-estimators. Here we model the residuals using IVW (i.e. intercept-free linear regression).

Although ultimately arbitrary, we focused on IVW, rather than another type of MR estimator, because it provides the most intuitive way to understand validity conditions: using IVW to create residuals means that pleiotropic effects in the residual creation step are passed forwards to the MR analysis. Hence, the estimator should produce valid estimates if a plurality of SNP effects are valid instruments. On the other hand, if weighted median was used in the first step then this would require that at least 50% of these variants would be valid. It is not obvious how the identification assumptions for the two steps would interact when defining which settings the estimator would be valid in. In addition, MBE are known to be much less precise than other estimators, and IVW is currently the most efficient multivariable estimator. Using other estimators to create residuals could exacerbate this issue.

Since the contamination mixture method has several advantages, discussed above, we also implemented this framework using both the contamination mixture method. This 'multivariable-CM' estimator uses IVW to create residuals fed into a contamination mixture model.

Our estimators are therefore algorithmic rather than model-based in the sense that we are not starting by precisely defining a statistical model, and then deriving conclusion from the assumptions of the model. But, instead, using an algorithm (taking the mode of the distribution) to convert genetic data in MR estimates. The likely trade-off for the conceptual simplicity of this approach will not optimise statistical efficiency.

**Deriving a standard error multivariable-MBE and multivariable-CM.** Assuming we have strong instruments (i.e. the first MR assumption is valid) we can use the first order approximation for the standard error of the Wald ratio that is typically used in two-sample MR studies. In a traditional univariable model this is defined as:

$$\mathrm{SE}_{\mathrm{wald},i} = \mathrm{SE}_{\mathrm{y},i}/|\beta_{x,i}| \qquad 5)$$

Where $\mathrm{SE}_{\mathrm{y},i}$ is the standard error of the $i$th variant-outcome association estimate.

In effect, this standard error is assuming that the variant-exposure association is measured with sufficient precision that we can assume that it contributes no error to the estimate of the causal effect. Under this assumption, the process of creating residuals will not increase the random error in the standard error of the Wald ratio. Hence, we model the standard error of the ith Wald ratio estimate as:

$$\mathrm{SE}_{\mathrm{resid},i} = \mathrm{SE}_{\mathrm{y},i}/|\beta_{x_1,i}| \qquad 6)$$

## Simulation study

We report our simulation study using the ADEMP (aims, data-generating mechanisms, estimands, methods, and performance measures) approach [21].

**Aims.**   We ran a simple simulation study to assess the performance of our plurality valid estimators when compared to other MVMR estimators.

**Data-generating mechanisms.**   We broadly simulate a setting in which there are two putative causal exposures for a single outcome. In the primary simulation we explore a setting in which the second exposure is pleiotropic (Fig 1), and where either both or neither of the exposures have a causal association with the outcome. We then explore how well the methods do under varying amounts of balanced and directional pleiotropy.

More formally, we simulated 200 single nucleotide polymorphisms (SNPs, which are common genetic variants) as independent and identically distributed binomial variables with the following parameters:

$$SNP \sim B(1, \ 0.4) + B(1, \ 0.4)$$

We additionally simulated the SNP effects on the exposures as independent and identically distributed normal variables

$$bSNP \sim N(0.05, \ 0.02^2)$$

The beta values and allele frequencies here were chosen to be loosely based on the effect sizes for the genome wide significant SNPs in the Wootton et al. UK Biobank GWAS smoking [22].

For settings in which we simulated pleiotropy (Fig 1.2A and 1.2B), the pleiotropic SNP effects were simulated as:

$$pSNP \sim N(BETA, \ SE^2)$$

Each simulation was repeated with BETA being set to either 0 or -0.03 to represent balanced and directional pleiotropy respectively. SE was always set to 0.1.

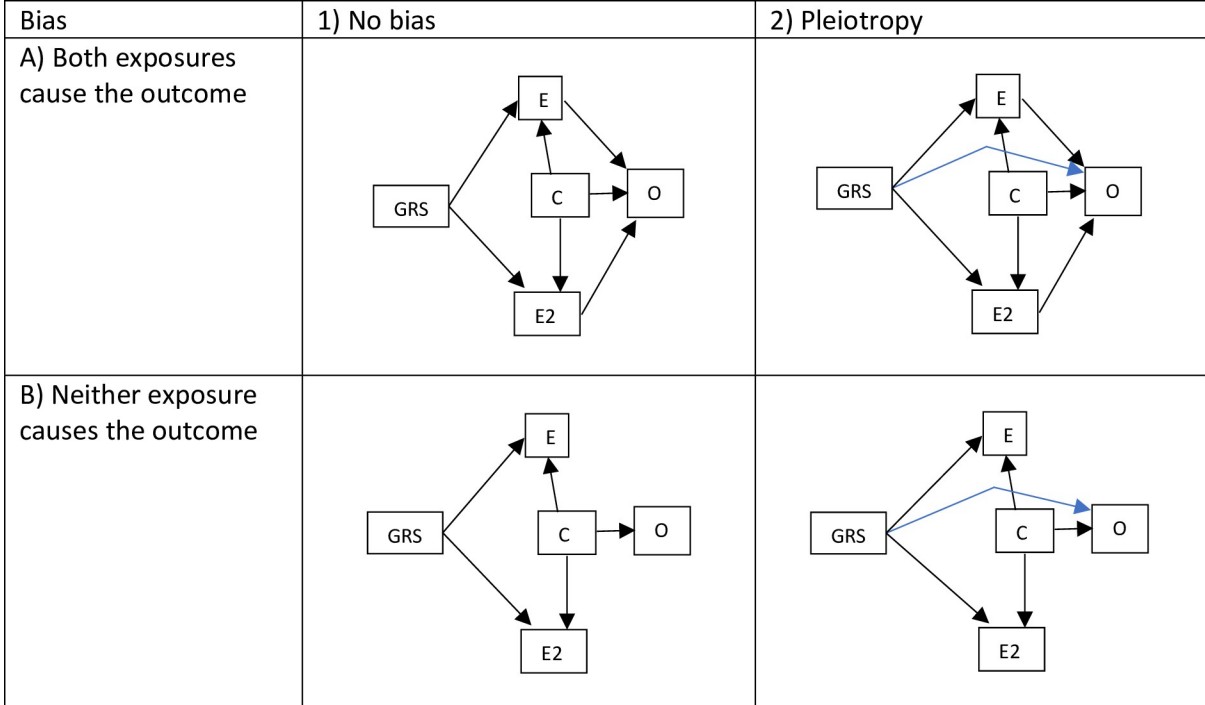

**Fig 1. Directed acyclic graphs of the simulation data generative models.** E and E2 are the first and second exposures respectively, GRS is the genetic liability to the exposures, and O is the outcome, and C is a confounder.

We then simulated a confounder as a normally distributed variable with the following parameters: $C \sim N(0, 1^2)$

We then defined the first exposure as:

$$E_1 = 0.3 * C + \sum_1^{200} [bSNP*SNP] + \varepsilon_1$$

where $\varepsilon$ is an error term such that $\varepsilon_1 \sim N(0, 1^2)$.

The second exposure was defined as:

$$E_{Pl} = 0.4 * C + \sum_1^{200} [bSNP*SNP] + \varepsilon_3$$

where $\varepsilon$ is an error term such that $\varepsilon_3 \sim N(0, 1^2)$.

When both exposures had null effects on the outcome (Fig 1.1B and 1.2B), the outcome was defined as:

$$O_{N;p} = C + \sum_0^p [pSNP*SNP] + \varepsilon_4$$

where $\varepsilon$ is an error term such that $\varepsilon_4 \sim N(0, 1^2)$. p could take the value of 0, 20, 40, or 80 to represent pleiotropic effects for 0, 10%, 20% or 40% of SNPs.

When both exposures had non-null effects on the outcome (Fig 1.1A and 1.2A), the outcomes were defined as:

$$O_{E1,EPl;p} = C + 0.3 * E_1 + 0.4 * E_{Pl} + \sum_1^p [pSNP**SNP] + \varepsilon_4$$

Where p takes the same definition as it had for $O_{N;p}$.

The phenotypic beta values chosen were chosen arbitrarily. However, biases are often more visible with larger effect estimates. By choosing realistically large betas we hoped to clearly illustrate the possible strengths and limitations of the different methods. While the specific results of our simulation may not be applicable to any specific applied setting, more general trends should be.

GWAS summary statistics for each exposure variable were estimated from linear regression models. Each genetic association with each exposure, and the outcome, were estimated from a unique sample of 200,000 participants with no sample overlap with the other GWASs.

**Estimands.** The causal effects of each exposure on the outcome.

*Methods*. We compare five methods for estimating the causal effect of the exposure on the outcome: multivariable IVW (intercept free multiple regression of the variant-outcome associations on the variant-exposure associations weighted by the inverse variance in the variant-outcome association), multivariable MR-Egger (multiple regression of the variant-outcome associations on the variant-exposure associations weighted by the inverse variance in the variant-outcome association), multivariable Weighted Median (quantile regression of the variant-outcome associations on the variant-exposure associations weighted by the inverse variance in the variant-outcome association), multivariable-MBE (using IVW to create the residuals and an MBE to estimate the causal effect), and multivariable-CM (using IVW to create the residuals and the contamination mixture method estimate the causal effect). IVW, MR-Egger, and weighted median were chosen because they appear to be some of the most widely used estimators which use different assumptions.

**Performance measure.** The primary performance measures were mean bias, 95% CI width, and the percentage of times that the confidence intervals include zero. When there is no causal effect, the latter will represent the type-2 error rate. When there is a causal effect, it measures one minus the type-1 error rate. In additional analyses we also explore the standard deviation of the effect estimate (overall 1000 simulations), and coverage for the causal effect of the

exposure on the outcome over the 1000 iterations. Bias was defined as the estimate minus the true causal effect. Thus, in the null settings, bias was the effect estimate. In the non-null settings, bias was the effect estimate of $E_1$ minus 0.3 and the estimate of $E_{Pl}$ minus 0.4. Coverage was defied as the percentage of times that the 95% confidence interval included the causal effect (or zero). 95% CI width was operationalised as difference between the upper 95% CI limit and the lower 95% CI limit.

## Applied example

We re-analysed the applied example (on the effect of intelligence, education, and household income on Alzheimer's disease) from Grant and Burgess' (2021) paper on pleiotropy robust estimators for MMVR [23]. This had previously been studied by Davies et al. and Anderson et al. [24, 25]. Anderson et al., in particular, had shown that a multivariable model was important for accounting for the collinearity between intelligence and education. Grant and Burgess then added household income to explore how the models worked with an additional risk factor.

Here we re-analysed the data used by Grant and Burgess (2021). They used 213 genetic variants from Davies et al. as instruments. These instruments had been clumped to ensure independence from each other and all had F statistics greater than 10, although the mean conditional F statistics ranged between 1.5 and 2.5. They used the Hill et al. GWAS of intelligence (n = 199,242 male and female European ancestry individuals) [26], Okbay et al. GWAS of years of education (n = 293,723 male and female European ancestry individuals) [27], and the Neale Lab UK Biobank GWAS of household income (n = 337,199 male and female European ancestry individuals) as sources of exposure data [28]. Since household income is an ordinal categorical variable, the genetic variant associations represent the increase in log odds of being in a higher income category per extra effect allele. Grant and Burgess (2021) additionally used Lambert et al. as a source of Alzheimer's data (n = 74,046 male and female European ancestry individuals) [29]. More information on the data sources can be found in the original publications.

We implemented our two novel estimators, as well as IVW, MR-Egger, and MR-Median. Since the genetic associations with education and intelligence were in the same direction, the MR-Egger estimates can be interpreted as being oriented in the direction of either of these exposures.

## Results

### Simulation

Table 1 presents the results for the primary performance measures (bias and 95% CI width) of the simulations from the settings in which both exposures cause the outcome, while in Table 2 neither exposure exerts a causal effect on the outcome. The mean conditional F statistic for Exposure 1 was around 197, and 186 for Exposure 2.

**Bias.** In both Tables 1 and 2, all estimators performed well in the no-bias setting. The small amount of bias observed (0.1% - 0.5%) is explicable by weak instrument bias and the variability in the estimates (S1 and S2 Tables). When there was balanced pleiotropy, the multivariable-MBE seemed to underperform the non-plurality valid estimators while the multivariable-CM estimator appeared to do slightly better. Multivariable-CM was comparatively unbiased by even large amounts of balanced pleiotropy. However, moderate amounts of directional pleiotropy were sufficient to bias estimates more than the Median estimator. For example, in the setting where both exposures are causal and there was 40% directional pleiotropy, the first and second exposure estimates were biased by -0.055 and -0.008 respectively for the Median estimator, but 0.073 and 0.054 for multivariable-CM. Multivariable-MBE was more biased than

**Table 1. Primary results for setting where both exposures cause the outcome, and exposure 2 is pleiotropic.**

| | | | No bias | 10% balanced pleiotropy | 20% balanced pleiotropy | 40% balanced pleiotropy | 10% directional pleiotropy | 20% directional pleiotropy | 40% directional pleiotropy |
|---|---|---|---|---|---|---|---|---|---|
| Bias | IVW | Exposure 1 | 0.001 | -0.003 | 0.003 | 0.001 | -0.026 | -0.048 | -0.109 |
| | | Exposure 2 | -0.006 | -0.002 | -0.008 | -0.004 | -0.034 | -0.067 | -0.121 |
| | MR Egger | Exposure 1 | 0.06 | 0.056 | 0.061 | 0.059 | 0.053 | 0.059 | 0.045 |
| | | Exposure 2 | -0.051 | -0.047 | -0.046 | -0.048 | -0.05 | -0.056 | -0.055 |
| | Median | Exposure 1 | 0.006 | 0.007 | 0.006 | 0.009 | 0.003 | 0.001 | -0.008 |
| | | Exposure 2 | -0.012 | -0.012 | -0.012 | -0.014 | -0.015 | -0.02 | -0.031 |
| | multivariable-CM | Exposure 1 | 0.003 | 0.002 | 0.003 | 0.004 | 0.006 | 0.015 | 0.073 |
| | | Exposure 2 | -0.005 | -0.006 | -0.007 | -0.007 | -0.005 | 0.001 | 0.054 |
| | multivariable-MBE | Exposure 1 | -0.001 | 0.001 | -0.079 | 0.149 | -0.055 | 0.154 | 0.253 |
| | | Exposure 2 | 0.074 | -0.049 | 0.179 | 0.598 | -0.117 | 0.054 | -0.113 |
| 95% CI width | IVW | Exposure 1 | 0.096 | 0.328 | 0.459 | 0.641 | 0.339 | 0.473 | 0.662 |
| | | Exposure 2 | 0.096 | 0.328 | 0.459 | 0.641 | 0.339 | 0.473 | 0.662 |
| | MR Egger | Exposure 1 | 0.172 | 0.464 | 0.638 | 0.885 | 0.478 | 0.656 | 0.91 |
| | | Exposure 2 | 0.173 | 0.464 | 0.637 | 0.883 | 0.477 | 0.655 | 0.908 |
| | Median | Exposure 1 | 0.14 | 0.147 | 0.156 | 0.18 | 0.147 | 0.155 | 0.178 |
| | | Exposure 2 | 0.14 | 0.148 | 0.157 | 0.18 | 0.147 | 0.156 | 0.179 |
| | multivariable-CM | Exposure 1 | 0.093 | 0.102 | 0.114 | 0.144 | 0.107 | 0.131 | 0.209 |
| | | Exposure 2 | 0.092 | 0.102 | 0.114 | 0.145 | 0.105 | 0.127 | 0.206 |
| | multivariable-MBE | Exposure 1 | 1.09 | 1.752 | 2.38 | 4.061 | 2.516 | 2.732 | 3.961 |
| | | Exposure 2 | 1.607 | 2.117 | 2.722 | 5.567 | 2.754 | 4.672 | 4.502 |
| % of times the 95% CI includes 0 | IVW | Exposure 1 | 0% | 0% | 0% | 0% | 0% | 0% | 0.5% |
| | | Exposure 2 | 0% | 0% | 0% | 0% | 0% | 0% | 0.1% |
| | MR Egger | Exposure 1 | 0% | 0% | 0% | 2.2% | 0% | 0% | 2.6% |
| | | Exposure 2 | 0% | 0% | 0% | 0.9% | 0% | 0% | 0.7% |
| | Median | Exposure 1 | 0% | 0% | 0% | 0% | 0% | 0% | 0% |
| | | Exposure 2 | 0% | 0% | 0% | 0% | 0% | 0% | 0% |
| | multivariable-CM | Exposure 1 | 0% | 0% | 0% | 0% | 0% | 0% | 0% |
| | | Exposure 2 | 0% | 0% | 0% | 0% | 0% | 0% | 0% |
| | multivariable-MBE | Exposure 1 | 6.4% | 13.3% | 22% | 31.9% | 15% | 24.6% | 35.5% |
| | | Exposure 2 | 7.5% | 14.5% | 21.6% | 33.8% | 13.8% | 20.5% | 35% |

multivariable-CM in all settings. For example, using the same simulation as above, multivariable-MBE was biased by 0.253 and -0.113 in the estimates for exposure 1 and 2 respectively.

**95% CI width.** The multivariable-MBE had the widest 95% CIs of all the estimators. For example, in the no bias simulation, the 95% CI widths were five to ten time larger than for the other estimators. The non-plurality valid estimators generally had similarly wide 95% CI. Multivariable-CM generally had tighter 95% CI than the other estimators.

**Coverage and power.** Since it had wide 95% CI, multivariable-MBE unsurprisingly had a low type-1 error rate (the 95% CI included the null in all settings > 98% when there was no association), but a high type-2 (the 95% CI included the null up to 35% of the time in settings where there was a true association). Multivariable-CM conversely had a very low type-2 error rate (the 95% CI never included the null when there was a true association). Multivariable-CM had a type-1 error rate at the nominal level (5%) for the 0% and 10% balance pleiotropy scenarios. In contrast, the Median estimator had type-1 error rates well below the nominal level in these scenarios. The type-1 error rates for Multivariable-CM were above the nominal level from 20% balanced pleiotropy, and for all levels of directional pleiotropy.

**Table 2. Primary results for setting where neither exposure causes the outcome, and exposure 2 is pleiotropic.**

| | | | No bias | 10% balanced pleiotropy | 20% balanced pleiotropy | 40% balanced pleiotropy | 10% directional pleiotropy | 20% directional pleiotropy | 40% directional pleiotropy |
|---|---|---|---|---|---|---|---|---|---|
| Bias | IVW | Exposure 1 | 0 | 0.004 | -0.002 | 0.003 | -0.029 | -0.056 | -0.111 |
| | | Exposure 2 | 0 | -0.004 | 0.002 | -0.001 | -0.03 | -0.059 | -0.116 |
| | MR Egger | Exposure 1 | 0 | 0.002 | 0.001 | 0.006 | -0.006 | -0.015 | -0.022 |
| | | Exposure 2 | 0 | -0.005 | 0.003 | 0.002 | 0 | -0.003 | -0.007 |
| | Median | Exposure 1 | 0 | 0 | 0 | 0.001 | -0.001 | -0.002 | -0.007 |
| | | Exposure 2 | 0 | 0 | 0 | 0 | -0.001 | -0.003 | -0.007 |
| | multivariable-CM | Exposure 1 | 0 | 0 | 0 | -0.001 | 0.009 | 0.047 | 0.166 |
| | | Exposure 2 | 0 | 0 | 0 | -0.001 | 0.009 | 0.046 | 0.167 |
| | multivariable-MBE | Exposure 1 | 0.004 | -0.062 | -0.168 | -0.276 | 0.097 | -0.17 | 0.208 |
| | | Exposure 2 | 0.008 | 0.02 | 0.12 | -0.062 | -0.117 | 0.189 | 0.903 |
| 95% CI width | IVW | Exposure 1 | 0.032 | 0.318 | 0.451 | 0.638 | 0.332 | 0.469 | 0.658 |
| | | Exposure 2 | 0.032 | 0.317 | 0.451 | 0.637 | 0.332 | 0.468 | 0.658 |
| | MR Egger | Exposure 1 | 0.044 | 0.435 | 0.618 | 0.873 | 0.454 | 0.641 | 0.9 |
| | | Exposure 2 | 0.044 | 0.434 | 0.616 | 0.871 | 0.453 | 0.64 | 0.898 |
| | Median | Exposure 1 | 0.046 | 0.051 | 0.057 | 0.071 | 0.052 | 0.057 | 0.073 |
| | | Exposure 2 | 0.046 | 0.051 | 0.057 | 0.071 | 0.052 | 0.057 | 0.073 |
| | multivariable-CM | Exposure 1 | 0.033 | 0.041 | 0.051 | 0.075 | 0.050 | 0.078 | 0.147 |
| | | Exposure 2 | 0.033 | 0.041 | 0.051 | 0.074 | 0.051 | 0.080 | 0.147 |
| | multivariable-MBE | Exposure 1 | 0.324 | 1.146 | 1.743 | 2.888 | 1.326 | 2.083 | 3.375 |
| | | Exposure 2 | 0.433 | 2.058 | 2.078 | 4.628 | 1.380 | 2.120 | 5.276 |
| % of times the 95% CI includes 0 | IVW | Exposure 1 | 95.7% | 95.6% | 94.7% | 95.7% | 93.5% | 93.1% | 90.2% |
| | | Exposure 2 | 94.6% | 96.2% | 95.1% | 94.5% | 92.2% | 92% | 88.3% |
| | MR Egger | Exposure 1 | 94.8% | 95.3% | 94.5% | 95.4% | 94% | 96.3% | 95% |
| | | Exposure 2 | 95.1% | 94.9% | 94.4% | 94.7% | 94.4% | 95.8% | 94.8% |
| | Median | Exposure 1 | 97.1% | 98% | 95.7% | 91.9% | 96.8% | 96.5% | 89.1% |
| | | Exposure 2 | 97.2% | 96.3% | 95.4% | 91.1% | 97.2% | 95.9% | 90.7% |
| | multivariable-CM | Exposure 1 | 95.4% | 95.3% | 92.1% | 86.1% | 89.7% | 63.3% | 23.5% |
| | | Exposure 2 | 94.7% | 95.1% | 91.7% | 85.4% | 88.1% | 65% | 24.7% |
| | multivariable-MBE | Exposure 1 | 99.4% | 99.2% | 99.2% | 99% | 99.2% | 98.7% | 98.9% |
| | | Exposure 2 | 99.7% | 99.5% | 99% | 98.9% | 99.1% | 99% | 98% |

**Additional outcomes.** Standard deviation of the effect estimates across the 1000 simulations: The SD of effect estimates between the multivariable-CM estimator and the non-plurality valid estimators were similar in the no-bias setting and when there was balanced pleiotropy (S1 and S2 Tables). However, multivariable-MBE had much wider SD, possibly because MBE produces less precise estimates than the contamination mixture method. In addition, all the plurality valid estimators had larger standard deviations when there was directional pleiotropy.

*Coverage.* Although all the estimators achieved 95% coverage when neither exposure was causal and there was no bias (S2 Table), surprisingly, except for Weighted Median and Multivariable-MBE, most estimators did not achieve at least 95% coverage when both exposures were causal (S1 Table). This might be because Weighted Median and Multivariable-MBE had the widest CI width (Tables 1 and 2) and all estimators were being effected by weak-instrument bias.

**Table 3. Results of the applied example exploring the effect to education and intelligence on Alzheimer's disease.**

| Method | Education (95% CI) | Intelligence (95% CI) | Household Income (95% CI) |
|---|---|---|---|
| IVW | -0.244 (-0.919 to 0.430) | -0.469 (-0.864 to -0.074) | 0.416 (-0.250 to 1.082) |
| Egger | -0.035 (-0.761 to 0.691) | -0.073 (-0.724 to 0.578) | 0.400 (-0.264 to 1.064) |
| Robust | -0.017 (-0.624 to 0.590) | -0.544 (-0.927 to -0.161) | 0.263 (-0.404 to 0.931) |
| Median | -0.134 (-0.873 to 0.606) | -0.573 (-1.029 to -0.116) | 0.368 (-0.378 to 1.114) |
| multivariable-CM | 0.046 (-0.601 to 0.689) | -0.575 (-0.920 to -0.198) | 0.303 (-0.288 to 0.893) |
| multivariable-MBE | 0.648 (-1.048 to 2.344) | -0.733 (-1.684 to 0.219) | 0.229 (-3.701 to 4.158) |

## Applied example

As with Grant and Burgess (2021), the pleiotropy robust estimators provided consistent estimates of the effects of education, intelligence, and household income on Alzheimer's disease (Table 3). All estimators concluded a null effect of education on Alzheimer's, conditional on the other exposures. However, they all implied a negative effect on intelligence, although the 95% CI for MR-Egger and multivariable-MBE included the null hypotheses. All estimators estimated a log odds ratio of household income around 0.3, but again with 95% CI which included zero. As the original study concluded "[t]he consistency of the findings give strength to the assertion that intelligence has a causally protective effect on Alzheimer's disease, conditional on years of education and household income. However, there is no evidence of a direct effect of years of education or household income on Alzheimer's disease."

## Discussion

Here we introduce two plurality valid estimators for multivariable Mendelian randomisation. Unlike most existing estimators, these use residual framework rather than multivariable regression models to produce the final effect estimates. We then used simulations with varying amounts of directional and balanced pleiotropy, as well as a re-analysis of the effect of intelligence, years of education, and household income on Alzheimer's disease to compare the relative performance of our estimators with each other and existing estimators for MVMR.

As with previous analyses, our estimators implied that intelligence has a protective effect on Alzheimer's disease, while years of education and household income do not. This has two important implications, firstly that as the years of mandatory education increase, there should not be a corresponding increase in Alzheimer's. Secondly, our results imply that public health interventions to boost intelligence, beyond additional years of education, may be useful in reducing the burden of Alzheimer's, although further research would be needed to confirm this hypothesis.

Of the two plurality valid estimators considered here, multivariable-CM, which uses IVW to create the residuals fed into a contamination mixture model, overall performed the best. It generally performed at least as well, if not better, than MR-Egger and IVW in terms of bias and precision in all settings. Indeed, when there was balanced pleiotropy, it was both more precise and less biased than IVW. However, in settings with moderate-to-high amounts of directional pleiotropy it was a lot more biased than Weighted median. Indeed, the high precision of the CM estimates is probably detrimental in this setting as it resulted in lower coverage than the other estimators. The divergence in performance between balanced and directional settings is probably, as discussed in the methods section, due to the multivariable-CM method assuming balanced pleiotropy. Hence, we would expect the estimator to perform better under situations where the distribution of Wald ratios with directional pleiotropy is similar to the assumed model with balanced pleiotropy, such as when the absolute amount of directional

bias is small. The MR-Egger intercept and funnel plots have both been suggested as methods for exploring the presence of directional pleiotropy, and therefore may be useful additional analyses when employing the multivariable-CM estimator [30]. Thus, while we think it can help triangulate results between a univariate and multivariable setting by allowing the use of a plurality valid estimator in both analyses, or between multiple multivariable estimators, we cannot recommend using it alone unless there is *a priori* evidence that there should be no directional pleiotropy.

Multivariable-MBE was sufficiently imprecise that it is likely to be uninformative in practice, and we would therefore suggest that, when needed, researchers use another robust multivariable method instead. The poorer performance of the MV-MBE estimator is probably due to the greater uncertainty in the estimates produced by the mode-based estimator [5]: in Tables 1 and 2, the bias remains meaningfully smaller than half of the 95% CI width, despite often being more than ten times greater than the bias for the other estimators.

Our simulations are not without limitations. Firstly, although pleiotropy can vary continuously between studies, we explore only discrete amounts of this biases. This could potentially mask non-linearities in the performance of pleiotropy robust estimators for MVMR in the presence of these biases. In addition, all our simulations assume linearity and homogeneity (i.e. no effect modification or interaction) of the effects of the risk factors on the outcomes. A further limitation of this work is that we have only considered the scenario with two exposures in our simulation study. However, the framework we introduce in this paper does naturally extend to consider more than two exposures by using multivariable IVW in the first stage. Finally, although multivariable-CM and multivariable-MBE could be implemented using estimates other than IVW to create residuals, here we have implemented it explicitly using IVW because the interpretation of the validity assumption using the other estimators is unclear.

In summary, here we introduce a framework for implementing plurality valid estimators for multivariable Mendelian randomisation in the absence of modal regression. Of these, the multivariable-CM estimator, which uses IVW to create residuals then fed into a contamination mixture method, appeared to perform the best. Although it performed very well with large amounts of balanced pleiotropy, it underperformed estimators like Weighted median when there was directional pleiotropy. We hope these estimators (available from https://github.com/bar-woolf/MVMRmode/wiki) will further enable the future triangulation of univariable MR studies which have used plurality valid estimators with multivariable MR designs.

## Supporting information

**S1 Table. Results for additional outcomes when both exposures cause the outcome, and exposure 2 is pleiotropic.**
(DOCX)

**S2 Table. Results for additional outcomes when neither exposure cause the outcome, and exposure 2 is pleiotropic.**
(DOCX)

## Acknowledgments

This work was carried out using the computational facilities of the Advanced Computing Research Centre, University of Bristol - http://www.bris.ac.uk/acrc/.

## Author Contributions

**Conceptualization:** Benjamin Woolf, Dipender Gill, Stephen Burgess.

**Methodology:** Benjamin Woolf, Andrew J. Grant, Stephen Burgess.

**Software:** Benjamin Woolf.

**Supervision:** Dipender Gill, Andrew J. Grant, Stephen Burgess.

**Writing – original draft:** Benjamin Woolf.

**Writing – review & editing:** Benjamin Woolf, Dipender Gill, Andrew J. Grant, Stephen Burgess.

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
