## [Decision Letter · Decision Letter 0]

5 May 2023

PONE-D-23-08704MVMRmode: Introducing an R package for plurality valid estimators for multivariable Mendelian randomisationPLOS ONE

Dear Dr. Woolf,

Thank you for submitting your manuscript to PLOS ONE. After careful consideration, we feel that it has merit but does not fully meet PLOS ONE’s publication criteria as it currently stands. Therefore, we invite you to submit a revised version of the manuscript that addresses the points raised during the review process. Please address the points raised by the reviewers one by one, particularly provide more details about the simulations, and discuss the comparision results with other methods in detail. Please submit your revised manuscript by Jun 19 2023 11:59PM. If you will need more time than this to complete your revisions, please reply to this message or contact the journal office at plosone@plos.org. Please include the following items when submitting your revised manuscript:A rebuttal letter that responds to each point raised by the academic editor and reviewer(s). You should upload this letter as a separate file labeled 'Response to Reviewers'.A marked-up copy of your manuscript that highlights changes made to the original version. You should upload this as a separate file labeled 'Revised Manuscript with Track Changes'.An unmarked version of your revised paper without tracked changes. You should upload this as a separate file labeled 'Manuscript'.

We look forward to receiving your revised manuscript.

Kind regards,

Suyan Tian

Academic Editor

PLOS ONE

Journal Requirements:

Reviewers' comments:

Reviewer's Responses to Questions

**Comments to the Author**

1. Is the manuscript technically sound, and do the data support the conclusions?

Reviewer #1: Yes

Reviewer #2: No

2. Has the statistical analysis been performed appropriately and rigorously? 

Reviewer #1: Yes

Reviewer #2: No

3. Have the authors made all data underlying the findings in their manuscript fully available?

Reviewer #1: Yes

Reviewer #2: No

4. Is the manuscript presented in an intelligible fashion and written in standard English?

Reviewer #1: Yes

Reviewer #2: No

5. Review Comments to the Author

Reviewer #1: This paper describes the application of the Mode estimator in the multivariable MR setting. The design of the study includes a simulation study and calculation of effects and comparison with effects of a previously published study on Alzheimer’s disease. The conclusions of the paper are supported by the application of the two mode estimators (multivariable-MBE and multivariable-CM) in the two studies. These findings show that while multivariable-MBE is of no use, the multivariable-CM outperforms other multivariable MR pleiotropy-robust methods such as the Egger and the Weighed median when balanced pleiotropy is present, but not when directional pleiotropy is present. The paper is well-written and I have no major comments.

My minor comments are the following:

Abstract: the CM stands for contamination mixture, please define this abbreviation

There is a typo in the conclusions: “to provided” instead of “to provide”

In the introduction of the abstract, there could be a mention on the fact that the modal method was developed as a complementary method to address pleiotropy in MR.

Can the term casual association be defined? Do the authors just mean causal association?

Could the authors better explain , or rephrase the following sentence: “we focused on IVW, rather than another type of MR estimator, because it provides the most intuitive to understand validity conditions”

In the data generating mechanisms of the ADEMP, there is a typo (casual instead of causal association).

In the methods section of the ADEMP definition, the authors compare the multivariable-MBE and multivariable-CM to the IVW, Egger and weighted median, but they do not specify if the latter 3 methods are applied in a univariate or multivariable MR framework.

In the performance measure paragraph, there is another typo (casual instead of causal)

Results section:

Did the authors expect the significant differences in performance metrics between the multivariable-MBE and the multivariable-CM? Are these differences comparable to those seen when the two mode estimands are applied in a univariate MR setting?

In the discussion of the paper, the message is clear and offers a guidance to the reader in terms of which method is the best to use in a given setting (ie presence of balanced vs directional pleiotropy). Could the authors also remind the reader how to evaluate empirically the presence of balanced vs directional pleiotropy in order to make the best choice of method for the multivariable MR?

Reviewer #2: The paper introduced two multivariable modal estimators though residual method for multivariable-MR, namely mulitivariable-MBE and multivariable-CM and developed a R package for this method. It provided a relatively comprehensive motivation for its proposal as they focused on the mode-based regression and plurality valid estimator for MVMR. However, some questions and concerns are listed as follows.

1. The description of the proposed method and its performance could be improved. There are typos and ambiguous terms. For example:

i). model assumptions and notations might be trivial but still should be included in the ‘Methods’ section before introducing equation 1) for clearer explanation on the proposed operations on estimates beta, alpha, etc.

ii). In ‘Simulation study’ section, capital P is used twice without introducing. I think the first P is a type but the second P in `B(1, P)` in the formula of outcome with non-null effects might be P / 100?

2. More theoretical details are needed or more careful discussion at least for the development and performance of the proposed methods. It seems more like a framework than a rigorous method to me and its performance closely related to the approach used to create the residuals and the precision of estimates. It’s good to discuss a little about why IVW method is the focus one though it’s more based on the interpretation perspective.

3. For the simulation results,

i). it may be better if the discussion of the choice of magnitude of coefficients and noise level are included as it will be interesting to see how other terms can affect the performance of this framework as no theoretical details are provided.

ii). Measurement metrics should be described, e.g. how bias is calculated in the table?

iii). More discussion or exploration should be provided. Based on the simulation and real data results and authors’ conclusion, multivariable-MBE generally provides more biased estimates than other methods and much larger but why? Is it due to the way residuals generated or estimates or the framework itself? This might also related to (2) the theoretical details or more comprehensive discussion of the method.

6. PLOS authors have the option to publish the peer review history of their article (what does this mean?). If published, this will include your full peer review and any attached files.

Reviewer #1: **Yes: **Despoina Manousaki

Reviewer #2: No

---

## [Author Response · Author response to Decision Letter 0]

3 Jul 2023

Suyan Tian 

Academic Editor 

PLOS ONE

Dear Dr Tian,

We would like to thank you and the reviewers for taking the time to assess our article “MVMRmode: Introducing an R package for plurality valid estimators for multivariable Mendelian randomisation" and providing feedback. Please find bellow our point-by-point response to the comments. We hope we now have adequately addressed all issues. a

Yours, 

Benjamin Woolf

Reviewer #1: 

“The paper is well-written and I have no major comments.”

Thank you very much!

My minor comments are the following:

“Abstract: the CM stands for contamination mixture, please define this abbreviation”

We have changed the sentence to read: 

multivariable-MBE, which uses IVW to create residuals fed into a traditional plurality valid estimator, and a method which instead has the residuals fed into the contamination mixture method (CM), multivariable-CM.

“There is a typo in the conclusions: “to provided” instead of “to provide””

Thank you, we have made the suggested correction. 

“In the introduction of the abstract, there could be a mention on the fact that the modal method was developed as a complementary method to address pleiotropy in MR.”

We have changed the second sentence of the abstract to read: 

Mode-based estimator (MBE) are one of the most popular types of estimators used in univariable-MR studies and is often used as a sensitivity analysis for pleiotropy.

“Can the term casual association be defined? Do the authors just mean causal association?”

“In the data generating mechanisms of the ADEMP, there is a typo (casual instead of causal association).”

“In the performance measure paragraph, there is another typo (casual instead of causal)”

Apologies for this typo, we’ve changed all references of ‘casual’ in the text to ‘causal’

“Could the authors better explain, or rephrase the following sentence: “we focused on IVW, rather than another type of MR estimator, because it provides the most intuitive to understand validity conditions” 

We have expanded the sentence to read: 

Although ultimately arbitrary, we focused on IVW, rather than another type of MR estimator, because it provides the most intuitive to understand validity conditions: using IVW to create residuals means that pleiotropic effects in the residual creation step are passed forwards to the MR analysis. Hence, the method should produce valid estimates if a plurality of SNP effects are valid instruments. On the other hand, if weighted median was used in the first step then this would require that at least 50% of these variants would be valid. It is not obvious how the identification assumptions for the two steps would interact when defining which settings the method would be valid in. In addition, MBE are known to be much less precise than other estimators, and IVW is currently the most efficient multivariable estimator. Using other estimators to create residuals could exacerbate this issue.

“In the methods section of the ADEMP definition, the authors compare the multivariable-MBE and multivariable-CM to the IVW, Egger and weighted median, but they do not specify if the latter 3 methods are applied in a univariate or multivariable MR framework.”

Sorry – these were all the multivariable implementation of the relevant methods. We have updated the text to read: 

We compare five methods for estimating the causal effect of the exposure on the outcome: multivariable IVW (intercept free multiple regression of the variant-outcome associations on the variant-exposure associations weighted by the inverse variance in the variant-outcome association), multivariable MR-Egger (multiple regression of the variant-outcome associations on the variant-exposure associations weighted by the inverse variance in the variant-outcome association), multivariable Weighted Median (quantile regression of the variant-outcome associations on the variant-exposure associations weighted by the inverse variance in the variant-outcome association), multivariable-MBE (using IVW to create the residuals and an MBE to estimate the causal effect), and multivariable-CM (using IVW to create the residuals and the contamination mixture method estimate the causal effect). IVW, MR-Egger, and weighted median were chosen because they appear to be some of the most widely used estimators which use different assumptions. 

“Did the authors expect the significant differences in performance metrics between the multivariable-MBE and the multivariable-CM? Are these differences comparable to those seen when the two mode estimand are applied in a univariate MR setting?”

We did not go into the study with a strong expectation of how the different methods would perform. As we note in the introduction, CM is more precise than most MBE estimators, so we did expect that CM would produce more precise estimates. 

“Could the authors also remind the reader how to evaluate empirically the presence of balanced vs directional pleiotropy in order to make the best choice of method for the multivariable MR?”

We have added the following to the discussion: 

The MR-Egger intercept and funnel plots have both been suggested as methods for exploring the presence of directional pleiotropy, and therefore may be useful additional analyses when employing the multivariable-CM estimator (27).

Reviewer #2: 

“1. The description of the proposed method and its performance could be improved. There are typos and ambiguous terms. For example:

i). model assumptions and notations might be trivial but still should be included in the ‘Methods’ section before introducing equation 1) for clearer explanation on the proposed operations on estimates beta, alpha, etc.”

We have added a "Notation and assumptions" section at the start of the ‘Theoretical background’ section which introduces the notation used in the read of this section. And, then kept the rest under a ‘statistical framework’ sub-section. The updated text can be found at the end of this letter. 

“ii). In ‘Simulation study’ section, capital P is used twice without introducing. I think the first P is a type but the second P in `B(1, P)` in the formula of outcome with non-null effects might be P / 100?”

Thank you for picking up on this. Yes, they are the same P and the second reference should have been P/100. We have added the following after the definition of OE1,EPl;P:

Where P takes the same definition as it had for ON;P. 

We hope this is clearer.

“2. More theoretical details are needed or more careful discussion at least for the development and performance of the proposed methods. It seems more like a framework than a rigorous method to me and its performance closely related to the approach used to create the residuals and the precision of estimates.”

Some statistical methods are model-based, whereas others are more algorithmic in nature. For example, clustering methods include Gaussian mixture modelling, which fits a formal statistical model for the datapoints, and nearest neighbour clustering, which is more algorithmic in nature. The latter approach is still a statistical method, in that it takes data inputs and provides cluster membership as outputs. In our case, we appreciate that this is not a model-based method, but it still takes data inputs and provides Mendelian randomization estimates are outputs. We note that algorithmic methods are more common in robust statistics, as the analyst does not always want to define the precise model followed by the data, but rather provide an estimate that performs reasonably for a variety of models. 

We have improved our exposition of the mode-based MVMR method. We have also clarified that this is an algorithmic method, rather than a model-based method at the end of the ‘statistical’ framework subsection: 

Our estimators are therefore algorithmic rather than model-based in the sense that we are not starting by precisely defining a statistical model, and then deriving conclusion from the assumptions of the model. But, instead, using an algorithm (taking the mode of the distribution) to convert genetic data in MR estimates. The likely trade-off for the conceptual simplicity of this approach will not optimise statistical efficiency. 

Finally, to avoid any contention, we have replaced most references to ‘method’ to ‘estimator’ or ‘framework’ where appropriate. 

“It’s good to discuss a little about why IVW method is the focus one though it’s more based on the interpretation perspective.”

We have expanded the discussion in the methods so that it now reads: 

Although ultimately arbitrary, we focused on IVW, rather than another type of MR estimator, because it provides the most intuitive to understand validity conditions: using IVW to create residuals means that pleiotropic effects in the residual creation step are passed forwards to the MR analysis. Hence, the method should produce valid estimates if a plurality of SNP effects are valid instruments. On the other hand, if weighted median was used in the first step then this would require that at least 50% of these variants would be valid. It is not obvious how the identification assumptions for the two steps would interact when defining which settings the method would be valid in. In addition, MBE are known to be much less precise than other estimators, and IVW is currently the most efficient multivariable estimator. Using other estimators to create residuals could exacerbate this issue.

“3. For the simulation results,

i). it may be better if the discussion of the choice of magnitude of coefficients and noise level are included as it will be interesting to see how other terms can affect the performance of this framework as no theoretical details are provided.”

We have added the following two sentences to the simulation methods section which we hope explains the logic behind the parameter choices:

The beta values and allele frequencies here were chosen to be loosely based on the effect sizes for the genome wide significant SNPs in the Woottan et al UK Biobank GWAS smoking (64).

The phenotypic beta values chosen were chosen arbitrarily. However, biases are often more visible with larger effect estimates. By choosing realistically large betas we hoped to clearly illustrate the possible strengths and limitations of the different methods. While the specific results of our simulation may not be applicable to any specific applied setting, more general trends should be. 

Finally, we had actually found when running small (e.g. 100 iteration) versions of our simulation that the results are actually reasonably robust to parameter choice, and certainly the trends observed here generalised to other parameterisations. However, we decided not to report this for two reasons: A) the computational time (about a week using an array job) to run the full simulation means that it is not feasible in practice to run the full simulation over many settings, and B) Since we already have three pages of tables with only one setting, it is unclear how to present the results in a way which is not overwhelming and is actually interpretable. 

“ii). Measurement metrics should be described, e.g. how bias is calculated in the table?”

We have added the following at the end of the performance measures subsection: 

Bias was defined as the estimate minus the true causal effect. Thus, in the null settings, bias was the effect estimate. In the non-null settings, bias was the effect estimate of E1 minus 0.3 and the estimate of EPl minus 0.4. Coverage was defied as the percentage of times that the 95% confidence interval included the causal effect (or zero). 95% CI width was operationalised as difference between the upper 95% CI limit and the lower 95% CI limit. 

We hope this is an adequate definition of the performance measures. 

“iii). More discussion or exploration should be provided. Based on the simulation and real data results and authors’ conclusion, multivariable-MBE generally provides more biased estimates than other methods and much larger but why? Is it due to the way residuals generated or estimates or the framework itself? This might also relate to (2) the theoretical details or more comprehensive discussion of the method.”

We suspect that it is due to the way that the residuals are created. Since MBEs are much less precise than other estimators, we suspect that part of the error might be accounted for by the much greater uncertainty in the estimates produced by the MBE used here. For example, even though the bias for MV-MBE is 10s or 100s of times larger than that for the other estimators in Table 1 and 2, it remains meaningfully smaller than half of the 95% CI width. We have added the following to the discussion:

The poorer performance of the MV-MBE estimator is probably due to the greater uncertainty in the estimates produced by the mode-based estimator: in Tables 1 and 2, the bias remains meaningfully smaller than half of the 95% CI width, despite often being more than ten times greater than the bias for the other estimators. 

 

##########################################################################

Notation and assumptions 

We assume a set of genetic variants that are independently distributed are being proposed as instruments in an MR analysis. We shall denote with subscript i the ith element of any vector, which relates to the ith genetic variant. Let β_(y,i) be the genetic variant-outcome association for the ith genetic variant and β_(x,i) be the genetic variant-exposure association for the ith variant. We represent the causal effect of the exposure on the outcome using the scalar θ. We also assume that the exposure-outcome relationship is linear and unaffected by effect modification. We let αi represent pleiotropic effects of the ith variant on the outcome. Thus, when αi = 0, the ith variant is a valid instrument.

Suppose the ith variant-exposure and variant-outcome associations are related according to the model proposed by Bowden et al. (16):

 β_(y,i)= θβ_(x,i)+ α_i

Now suppose we have estimates for two exposures, denoted by x_1 and x_2. β_(x_1,i) and β_(x_2,,i) are the ith variant’s associations with the first and second exposure, respectively. Likewise, θ_1 and θ_2 are the causal effects of the first and second exposure, respectively, on the outcome. We can now extend (1) as follows:

 β_(y,i)= θ_1 β_(x_1,i)+θ_2 β_(x_2,i)+α_i^'

Where α_i^' represents pleiotropic effects of the ith variant on the outcome which do not pass via x_1 or x_2.

Statistical framework

In practice, we do not observe β_y, β_(x_1 ), or β_(x_2 ). However, we may obtain estimates, for example from GWAS. We denote the vectors of association estimates by β ^_y β ^_(x_1 ), and β ^_(x_2 ). Thus, in traditional multivariable-IVW we can estimate θ_1and θ_2 using the following linear model: 

 β ^_y= θ_1 β ^_(x_1 )+θ_2 β ^_(x_2 )+ε_1; and ε_(1,i) ~ N(0,σ_(y,i)^2 ).

Given the data structure in equation (2), (3) will provide a consistent estimator when α_i^'=0 for all i (i.e., all variants are valid instruments), or when ∑_1^n▒α_i^' =0 and α_i^' is independent of β ^_(x_(1,i) ) and β ^_(x_(2,i) ) for all i (i.e., pleiotropy is balanced and the InSIDE assumption is met). A plurality valid estimator, on the other hand, should be consistent provided that a plurality of the α_i^' are zero, i.e. under the ZEMPA assumption.

Let 〖 β ~〗_y be the residuals from regressing β ^_y on β ^_(x_2 ) (without an intercept), and let 〖 β ~〗_x1be the residuals from regressing β ^_(x_1 ) on β ^_(x_2 ) (without an intercept). We can now estimate θ_1 using the linear model:

 〖 β ~〗_y= θ_1 〖 β ~〗_x1+ ε_2 ; and ε_(2,i) ~ N(0,σ_y^2 ).

Let α ~ be the residuals from regressing a vector of the pleiotropic effects on β ^_(x_2 ) (without an intercept). Because we have now reformulated the equation for the variant-outcome association so that it is in terms of a univariable regression model, 〖 β ~〗_y and 〖 β ~〗_x1 can be used as the inputs to a traditional univariable mode-based estimator. When more than one exposure is of interest, then this process can be iterated for each exposure. It follows that a plurality valid estimator for θ_1 using the residuals in this way will produce a valid estimate provided that a plurality of the α ~_i values are zero. This seems likely to be the case if a plurality of the α_i^' values are zero and the non-zero elements are distributed around zero (i.e., balanced pleiotropy). 

In settings with only two exposures, the residuals could be obtained through univariable MR of the outcome on the second exposure, and of the first exposure on the second exposure. Where there are more than two exposures, an existing multivariable MR method could be used instead to create residuals. This general framework could be implemented using a variety of estimators. Here we explore two types of plurality valid estimators. Firstly, we explore an estimator which uses a regression model to create the residuals fed into a traditional mode-based estimator (MBE) (8), which we dub ‘multivariable-MBE’. This regression model could be created using any of the existing MVMR-estimators. Here we model the residuals using IVW (i.e. intercept-free linear regression). 

Although ultimately arbitrary, we focused on IVW, rather than another type of MR estimator, because it provides the most intuitive to understand validity conditions: using IVW to create residuals means that pleiotropic effects in the residual creation step are passed forwards to the MR analysis. Hence, the estimator should produce valid estimates if a plurality of SNP effects are valid instruments. On the other hand, if weighted median was used in the first step then this would require that at least 50% of these variants would be valid. It is not obvious how the identification assumptions for the two steps would interact when defining which settings the estimator would be valid in. In addition, MBE are known to be much less precise than other estimators, and IVW is currently the most efficient multivariable estimator. Using other estimators to create residuals could exacerbate this issue. 

Since the contamination mixture method has several advantages, discussed above, we also implemented this framework using both the contamination mixture method. This ‘multivariable-CM’ estimator uses IVW to create residuals fed into a contamination mixture model. 

Our estimators are therefore algorithmic rather than model-based in the sense that we are not starting by precisely defining a statistical model, and then deriving conclusion from the assumptions of the model. But, instead, using an algorithm (taking the mode of the distribution) to convert genetic data in MR estimates. The likely trade-off for the conceptual simplicity of this approach will not optimise statistical efficiency.

---

## [Decision Letter · Decision Letter 1]

17 Jul 2023

PONE-D-23-08704R1MVMRmode: Introducing an R package for plurality valid estimators for multivariable Mendelian randomisationPLOS ONE

Dear Dr. Woolf,

Thank you for submitting your manuscript to PLOS ONE. After careful consideration, we feel that it has merit but does not fully meet PLOS ONE’s publication criteria as it currently stands. Therefore, we invite you to submit a revised version of the manuscript that addresses the points raised during the review process. Please correct the typos in the manuscript.

We look forward to receiving your revised manuscript.

Kind regards,

Suyan Tian

Academic Editor

PLOS ONE

Journal Requirements:

Reviewers' comments:

Reviewer's Responses to Questions

**Comments to the Author**

1. If the authors have adequately addressed your comments raised in a previous round of review and you feel that this manuscript is now acceptable for publication, you may indicate that here to bypass the “Comments to the Author” section, enter your conflict of interest statement in the “Confidential to Editor” section, and submit your "Accept" recommendation.

Reviewer #1: All comments have been addressed

Reviewer #2: All comments have been addressed

2. Is the manuscript technically sound, and do the data support the conclusions?

Reviewer #1: Yes

Reviewer #2: Yes

3. Has the statistical analysis been performed appropriately and rigorously? 

Reviewer #1: Yes

Reviewer #2: Yes

4. Have the authors made all data underlying the findings in their manuscript fully available?

Reviewer #1: Yes

Reviewer #2: Yes

5. Is the manuscript presented in an intelligible fashion and written in standard English?

Reviewer #1: Yes

Reviewer #2: Yes

6. Review Comments to the Author

Reviewer #1: I thank the authors for addressing my previous comments. I have no further suggestions, other than a small typo (intuitive (adjective) should be replaced by "intuition" or "intuitive way")

Reviewer #2: Thank you for taking your time to carefully address all the questions! All in all, this paper introduced two algorithmic methods for multivariable-MR, namely the multivariable-MBE and multivarible-CM. They adopted Monte-carlo simulations for the performance of the proposed estimators and adopted them to study the causal effect of intelligence, education and household income on Alzheimer’s disease for real data analysis, along with the existing methods. I think in this revised version, a much clearer explanation and description have been made for your proposed methods.

1. There are still some typos and unclear parts you might want to correct, for example:

(i) the 2,, in the sentence “Now suppose we have estimates for two exposures, denoted by 1 and 2. 1, and 2,,” right after equation (1).

(ii) In simulation part, it’s mentioned that 200 SNPs are generated and generation of E1, E_PI includes the first 50 and first 100 SNPs, respectively. Do you use all the 200 SNPs or only the first 100 ones? Also there is a small p in the definition of O_N;P, I think you mean p = P / 100 * 200, is that correct?

(iii) In the definition of O_E1, EPI;P, I think the 100 should be P or p?

(iv) ‘causals’ should be causal in the discussion of coverage as the additional outcomes in the Result section for simulation.

2. Will you conclude that multivariable-MBE is not that useful in practice? Or provide some suggestions for under what circumstance this method might work? Maybe an inclusion of the MV-MBE performance under different levels of pleiotropy can be used to better explain why it fail in this framework, or include the sd for the estimator to support the argument “probably due to the greater uncertainty in the estimates”.

3. Just for comprehensive analysis, a more detailed discussion for more than two exposures will be ideal I think, will complicate the analysis for sure though. Because I’m kind of curious how different methods in this framework affect the final results and this kind of analysis, e.g. some patterns shown in the simulation, might provide some insights for your framework as well.

7. PLOS authors have the option to publish the peer review history of their article (what does this mean?). If published, this will include your full peer review and any attached files.

Reviewer #1: **Yes: **Despoina Manousaki

Reviewer #2: No

---

## [Author Response · Author response to Decision Letter 1]

16 Aug 2023

Suyan Tian 

Academic Editor 

PLOS ONE

Dear Dr Tian,

We would like to thank you and the reviewer for again taking the time to assess our article "MVMRmode: Introducing an R package for plurality valid estimators for multivariable Mendelian randomisation”, and providing such generous feedback. I apologise for most of these errors being typos – I’m badly dyslexic and struggle to pick up this type of mistake. Please find below our point-by-point response to the comments bellow where we have implemented all of the suggested changes. 

Yours, 

Benjamin Woolf

Journal Requirements:

“Please review your reference list to ensure that it is complete and correct. If you have cited papers that have been retracted, please include the rationale for doing so in the manuscript text, or remove these references and replace them with relevant current references. Any changes to the reference list should be mentioned in the rebuttal letter that accompanies your revised manuscript. If you need to cite a retracted article, indicate the article’s retracted status in the References list and also include a citation and full reference for the retraction notice.”

To the best of our knowledge we have not cited any retracted papers. In the process of revising the manuscript, it has come to our attention that other plurality-valid estimators for multivariable Mendelian randomization (MVMR) have been published (either as a journal article or as a pre-print). We have amended the Introduction and Discussion to clarify that this is no longer the first plurality-valid MVMR method, and added reference to these methods.

Reviewers' comments

Reviewer #1: 

 “a small typo (intuitive (adjective) should be replaced by "intuition" or "intuitive way")”

Thank you for picking this up, we have changed “intuitive” to " intuitive way "

Reviewer #2: 

 “There are still some typos and unclear parts you might want to correct, for example: (i) the 2,, in the sentence “Now suppose we have estimates for two exposures, denoted by 1 and 2. 1, and 2,,” right after equation (1).”

Thank you for picking this up, I have removed the second comma form β_(x_2,i)

 “(ii) In simulation part, it’s mentioned that 200 SNPs are generated and generation of E1, E_PI includes the first 50 and first 100 SNPs, respectively. Do you use all the 200 SNPs or only the first 100 ones? Also there is a small p in the definition of O_N;P, I think you mean p = P / 100 * 200, is that correct?”

Thank you very much for picking this up. After double checking with the code, it should be 200 SNPs for both. I’m sorry for not having picked this up this version control issue before. I have changed the ‘P’ to a ‘p’. 

 “(iii) In the definition of O_E1, EPI;P, I think the 100 should be P or p?”

Thank you again for picking this up, it should be ‘p’. I have changed the ‘100’ to ‘p’

 “(iv) ‘causals’ should be causal in the discussion of coverage as the additional outcomes in the Result section for simulation.”

Thank you for picking these up, we have made the suggested changes. 

 “Will you conclude that multivariable-MBE is not that useful in practice? Or provide some suggestions for under what circumstance this method might work? Maybe an inclusion of the MV-MBE performance under different levels of pleiotropy can be used to better explain why it fail in this framework, or include the sd for the estimator to support the argument “probably due to the greater uncertainty in the estimates”.

We agree with the reviewer, and have clarified this in the text: 

Multivariable-MBE was sufficiently imprecise that it is likely to be uninformative in practice, and we would therefore suggest that, when needed, researchers use another robust multivariable method instead.

The claim “probably due to the greater uncertainty in the estimates” relates to the univariable mode-based method, which has been shown to have variable estimates in previous published comparisons of univariable Mendelian randomization methods. We now provide a citation for this claim.

 “Just for comprehensive analysis, a more detailed discussion for more than two exposures will be ideal I think, will complicate the analysis for sure though. Because I’m kind of curious how different methods in this framework affect the final results and this kind of analysis, e.g. some patterns shown in the simulation, might provide some insights for your framework as well.”

We agree with the reviewer that this is a limitation of the current manuscript. However, due to the limited performance of the proposed methods in the current simulation scenarios, and the recent publication of other plurality-robust methods, we did not this it would be interesting to readers to provide a comparison of these methods in a more complex scenario.

In the text we have added the following: 

“A further limitation of this work is that we have only considered the scenario with two exposures in our simulation study. However, the framework we introduce in this paper does naturally extend to consider more than two exposures by using multivariable IVW in the first stage.”

---

## [Editor Report · Decision Letter 2]

23 Aug 2023

MVMRmode: Introducing an R package for plurality valid estimators for multivariable Mendelian randomisation

PONE-D-23-08704R2

Dear Dr. Woolf,

We’re pleased to inform you that your manuscript has been judged scientifically suitable for publication and will be formally accepted for publication once it meets all outstanding technical requirements.

Kind regards,

Suyan Tian

Academic Editor

PLOS ONE

Additional Editor Comments (optional):

All points raised by the reviewers have been addressed appropriately.
---

## [Editor Report · Acceptance letter]

29 Aug 2023

PONE-D-23-08704R2 

MVMRmode: Introducing an R package for plurality valid estimators for multivariable Mendelian randomisation 

Dear Dr. Woolf:

I'm pleased to inform you that your manuscript has been deemed suitable for publication in PLOS ONE. Congratulations! Your manuscript is now with our production department. 

Kind regards, 

on behalf of

Dr. Suyan Tian 

Academic Editor

PLOS ONE